# The DUX4–HIF1α Axis in Murine and Human Muscle Cells: A Link More Complex Than Expected

**DOI:** 10.3390/ijms25063327

**Published:** 2024-03-15

**Authors:** Thuy-Hang Nguyen, Maelle Limpens, Sihame Bouhmidi, Lise Paprzycki, Alexandre Legrand, Anne-Emilie Declèves, Philipp Heher, Alexandra Belayew, Christopher R. S. Banerji, Peter S. Zammit, Alexandra Tassin

**Affiliations:** 1Laboratory of Respiratory Physiology, Pathophysiology and Rehabilitation, Research Institute for Health Sciences and Technology, University of Mons, 7000 Mons, Belgium; 2Department of Metabolic and Molecular Biochemistry, Research Institute for Health Sciences and Technology, University of Mons, 7000 Mons, Belgium; 3Randall Centre for Cell and Molecular Biophysics, King’s College London, Guy’s Campus, London SE1 1UL, UK; 4The Alan Turing Institute, The British Library, London NW1 2DB, UK

**Keywords:** FSHD, DUX4, HIF1α, myogenesis, skeletal muscle

## Abstract

FacioScapuloHumeral muscular Dystrophy (FSHD) is one of the most prevalent inherited muscle disorders and is linked to the inappropriate expression of the DUX4 transcription factor in skeletal muscles. The deregulated molecular network causing FSHD muscle dysfunction and pathology is not well understood. It has been shown that the hypoxia response factor HIF1α is critically disturbed in FSHD and has a major role in DUX4-induced cell death. In this study, we further explored the relationship between DUX4 and HIF1α. We found that the DUX4 and HIF1α link differed according to the stage of myogenic differentiation and was conserved between human and mouse muscle. Furthermore, we found that HIF1α knockdown in a mouse model of DUX4 local expression exacerbated DUX4-mediated muscle fibrosis. Our data indicate that the suggested role of HIF1α in DUX4 toxicity is complex and that targeting HIF1α might be challenging in the context of FSHD therapeutic approaches.

## 1. Introduction

FacioScapuloHumeral muscular Dystrophy (FSHD) is a dominant hereditary disease characterized by a progressive and often left/right asymmetric skeletal muscle weakness that initially affects facial muscles and progresses through a rostro-caudal pattern. FSHD lowers quality of life, and approximately 30% of patients become wheelchair-bound [1,2]. FSHD involves complex genetic and epigenetic components leading to the activation of *DUX4* in skeletal muscle, a gene that encodes a potent pioneer transcription factor [3,4,5,6,7,8]. DUX4 is normally expressed in germline and early embryogenesis, where it plays a role in zygotic genome activation and, in mouse, appears involved in placentation [9,10,11]. The FSHD epigenetic defect is located in the *4q35* chromosome region at the macrosatellite *D4Z4* repeat array, which is hypermethylated in unaffected individuals and therefore in a closed chromatin conformation. FSHD results from the hypomethylation and epigenetic derepression of the *D4Z4* repeat array and, thus, a more permissive chromatin structure, which allows *DUX4* gene transcription from the distal-most *D4Z4* unit [1,3,12]. *DUX4* RNAs extend from the distal *D4Z4* unit to the flanking *pLAM* region, where they acquire an intron and an exon with a polyadenylation signal (PAS), allowing for the production of a stable mRNA that can be translated to generate DUX4 protein. This PAS sequence is only present on some permissive distal *4qA* alleles but not on *4qB* [13,14]. 

DUX4 protein toxicity likely results from disturbances in many different cell signaling pathways. DUX4 expression is rare and likely then, only in short bursts. Even so, DUX4 transcriptional activity causes widespread gene deregulation activating germline-specific genes [5], but inhibiting genes involved in myogenesis [15] and altering the expression of genes involved in the oxidative stress response [16,17,18,19,20,21,22,23,24] (reviewed in [8]). DUX4 was found to not only trigger oxidative stress itself but also to regulate about 200 genes indirectly through oxidative stress [25]. Second, based on single-cell RNA sequencing (sc-RNAseq) and microarray data, Banerji and Zammit determined that the repression of a PAX7 target gene signature is a reliable biomarker of the pathology and is associated with disease progression [1,15,26]. PAX7 is a myogenic transcription factor that regulates satellite cells, the resident stem cells of skeletal muscle. The single PAX7 homeodomain is similar to the two of DUX4 [27], and PAX7 can rescue DUX4-induced cytotoxicity in mouse muscle cells [16]. DUX4 and PAX7 can bind similar DNA elements and compete on reporter target gene activation [15]. Third, several studies have documented a large discrepancy between the transcriptomic (RNAseq) and proteomic landscapes in DUX4-expressing muscle cells, highlighting disturbances of post-transcriptional processes such as RNA splicing and RNA and protein quality control pathways [28,29,30,31]. Finally, other studies have suggested that DUX4 interaction with specific protein partners can participate in FSHD-associated pathological processes, such as interference with regeneration [32,33]. 

The molecular map of FSHD-associated interaction signaling established by Banerji et al. [34] and based on a meta-analysis of microarray data sets from FSHD muscle biopsies demonstrated that the hypoxia response pathway was critically perturbed in FSHD, among other pathways such as the WNT pathway. In FSHD, PAX7 repression was also associated with the induction of hypoxia-response genes [15]. Accordingly, Tsumagari et al. independently described the HIF1α-signaling network as one of the over-represented pathways among FSHD dysregulated genes [35]. HIF1α is the master regulator of physiological adaptive mechanisms in response to hypoxia [36]; it induces the expression of multiple effector genes that modulate various cellular processes, such as glucose metabolism and oxidative stress. In skeletal muscle, hypoxia modulates not only the muscle fiber type profile but also myogenesis, regeneration and vascularization. The activation of HIF1α can also be involved in pathological conditions independently from hypoxia. This “pseudohypoxia” was mostly described in cancer [37]. In FSHD, available data suggest that the primary genetic defect can cause HIF1α pathway disturbances through a putative DUX4–HIF1α axis (as we reviewed in [38]). Indeed, HIF1α was identified as necessary for DUX4 toxicity by Lek et al. in a genome-wide CRISPR-Cas9 screen performed to identify genes whose loss of function allows survival of myoblasts expressing DUX4 [39]. However, the DUX4–HIF1α axis requires clarification, particularly taking into account variations in HIF1α expression and effects during myogenic differentiation [40].

If one considers a putative pathological contribution of HIF1α to FSHD, it is noteworthy that HIF1α promotes a metabolic switch in favor of anaerobic glycolysis, the metabolic pathway favored in early embryogenesis, where DUX4 and mouse Dux have a function in zygotic genome activation [9,10,41]. Interestingly, most stem cell types reside in hypoxic niches where HIF1α controls pluripotency gene expression, promotes glycolytic metabolism and inhibits mitochondrial biogenesis. Aberrant DUX4/HIF1α activation could therefore contribute to metabolic disturbances in adult FSHD muscle cells. In addition, HIF1α modulates myogenic differentiation [40], a process that occurs during skeletal muscle regeneration and is disturbed in FSHD [42,43]. DUX4-induced HIF1α pathway misregulation could therefore participate in the FSHD-associated defect in adult myogenesis. Accordingly, we showed that DUX4 suppresses the HIF1α-mediated precocious differentiation of human myoblasts [40]. Moreover, Heher et al. showed that environmental hypoxia aggravates the hypotrophic FSHD myotube phenotype. This effect is due to a DUX4-mediated metabolic mis-adaptation, leading to exacerbated oxidative stress through disproportionally enhanced reactive oxygen species (ROS) formation under low O_2_ tension [18]. 

From a therapeutic perspective, Lek et al. [39] showed that pharmacological HIF1α signaling inhibitors could improve DUX4-associated toxicity in myoblast cultures and muscle phenotypes in FSHD-like zebrafish embryos. However, this approach could prove deleterious at later muscle development stages because of the negative effect of indirect HIF1α inhibitors on protein synthesis or on HIF1α contribution to the myogenic program. In addition, the effects of a specific HIF1α knockdown have not been investigated in mature muscle nor in a murine model of *DUX4* expression. 

In summary, the HIF1 pathway is clearly disturbed in FSHD muscle, but discrepancies remain regarding HIF1α involvement in FSHD pathophysiological mechanisms. Lek et al. [39] reported that HIF1 pathway suppression prevented DUX4 toxicity, either by using CRISPR-Cas9 gene inactivation in a human myoblast cell line or by pharmacological inhibition in human myoblasts and in zebrafish embryos. HIF1 inhibitors developed in the field of cancer are thus considered a therapeutic strategy for FSHD. However, major questions remain, notably regarding DUX4-induced HIF1α pathway variations during the myogenic process or in mature muscle. Importantly, the relevance of HIF1α inhibitors as a therapeutic approach must be further interrogated, given (i) HIF1α function in the myogenic program, (ii) the metabolic shuffle occurring during this process and (iii) the specific and potentially deleterious effects of HIF1 inhibitors on skeletal muscle, notably on protein synthesis. The present study therefore aimed to fill the gap of knowledge regarding (i) the link between DUX4 and HIF1α during human muscle cell differentiation [44], (ii) its conservation in murine models of *DUX4* expression in vitro [16] and in vivo [45] and (iii) the effect of a targeted HIF1α knockdown on DUX4-mediated muscle lesions. 

## 2. Results

### 2.1. Effect of DUX4 on HIF1α Expression and Nuclear Protein Level at Different Stages of Human Myoblast Differentiation In Vitro

We used the LHCN-M2-iDUX4 cell line with doxycycline (DOX)-inducible *DUX4* expression, engineered from the immortalized human myoblast line LHCN-M2 [44,46]. Because of its high toxicity, the impact of DUX4 induction was first evaluated by cell viability assays (MTT and CCK8) under a standard oxygen partial pressure (21% PO_2_), 24 h after DOX addition to the culture medium. These tests were performed in LHCN-M2-iDUX4 myoblasts (Appendix A) as well as on LHCN-M2 to test DOX toxicity (Appendix A). DUX4 induction had no effect on cell viability up to 62.5 ng/mL DOX. From 125 ng/mL and above, a significant decrease in viability was observed in LHCN-M2-iDUX4 cells in both the MTT and CCK8 tests (Appendix A). DOX exposure had no effect on myoblast viability in the absence of *DUX4* expression, as verified in LHCN-M2 cells (Appendix A). The production of the DUX4 protein was also confirmed by immunofluorescence (IF) in LHCN-M2-iDUX4 myoblasts (Appendix A). After the addition of 15.6 or 31.3 ng/mL of DOX to the culture medium for 24 h, we detected 31.5% (±7.4%) and 55.6% (±9.8%) DUX4-positive (DUX4^+^) nuclei, respectively. This percentage rose to 80% with 62.5 and up to 250 ng/mL of DOX but was not significantly different among these higher doses. The dose of 62.5 ng/mL of DOX was therefore selected for further experiments because it showed the best DUX4 induction at a level with no effect on cell viability in the short term. 

The impact of DUX4 on HIF1α mRNA and protein level at a standard PO_2_ of 21% was then studied in LHCN-M2-iDUX4 cells in proliferation (myoblasts), as well as in early (myocytes) and late (myotubes) differentiation stages. 

In proliferating myoblasts, DUX4 induction nearly halved the proportion of HIF1α-positive (HIF1α^+^) nuclei with 43.9% (± 2.7%) HIF1α^+^ nuclei in control cells but only 25.9% (±4.5%) in the presence of DUX4 (Figure 1A–C and Appendix A). Accordingly, the *HIF1A* mRNA level was halved in proliferating myoblasts expressing DUX4 (Figure 1D), and only 17.8% of HIF1α^+^ nuclei presented DUX4 IF labeling (Figure 1E,F). 

In myocytes with DUX4 induction, the percentage of HIF1α^+^ nuclei trended toward a decrease without reaching statistical significance (*p* = 0.051, *t*-test) (Figure 1G–I and Appendix A), whereas the *HIF1A* mRNA level was not significantly changed (Figure 1J). At this stage, 8.1% of HIF1α^+^ nuclei presented DUX4 IF labeling (Figure 1K,L).

In myotubes, the percentage of HIF1α^+^ nuclei doubled upon DUX4 induction (31.3% ± 4.9%) compared to the uninduced controls (16.3% ± 3.5%) (Figure 1M–O and Appendix A). Concomitantly, the *HIF1A* mRNA level was significantly increased (Figure 1P). At this stage, 52.3% of HIF1α^+^ nuclei presented DUX4 IF labeling (Figure 1Q,R).

A comparison of HIF1α expression patterns in myoblasts, myocytes and myotubes (Appendix A) showed that the *HIF1α* mRNA level and the percentage of HIF1α^+^ nuclei decreased during the differentiation process in non-induced LHCN-M2-iDUX4 cells. Upon *DUX4* expression, this pattern was conserved at the mRNA level, but the percentage of HIF1α^+^ nuclei increased at the myotube stage as compared to myocytes.

### 2.2. Effect of DUX4 on HIF1α Target Genes at Different Stages of Human Myoblast Differentiation In Vitro

To further determine the impact of DUX4 on the HIF1α pathway at 21% PO_2_, we studied the effect of *DUX4* expression on two direct HIF1α transcriptional targets: *VEGFA* and *PDK1*. The *PDK1* mRNA level was decreased twofold in proliferating myoblasts expressing DUX4, but the *VEGFA* mRNA level was not significantly changed compared to non-induced cells (Figure 2A). In agreement with our results obtained at the RNA level, Western blotting showed a significant decrease in PDK1 protein abundance in proliferating myoblasts after the induction of *DUX4* expression (Figure 2B,C). Similarly, as observed at the mRNA level, an ELISA on the culture medium showed no change in the VEGF protein level (Figure 2D). In myocytes expressing DUX4, the mRNA level of both HIF1α target genes was not significantly changed (Figure 2E), but PDK1 and VEGF protein relative abundances were reduced (Figure 2F–H). In myotubes, similar to the results obtained for HIF1α, the mRNA levels of its target genes, *PDK1* and *VEGFA*, were significantly increased three- and four-fold, respectively (Figure 2I). However, the PDK1 (but not VEGF) protein level was significantly decreased upon *DUX4* expression (Figure 2J–L).

A comparison of HIF1α target gene expression in myoblasts, myocytes and myotubes (Appendix A) indicated a decrease in *PDK1* and *VEGFA* mRNA levels during the differentiation process in non-induced LHCN-M2-iDUX4 cells. On the contrary, upon *DUX4* expression, both *PDK1* and *VEGFA* mRNA levels showed a trend of increasing through differentiation. These kinetics were in accordance with the changes observed in the percentage of HIF1α^+^ nuclei.

### 2.3. Effect of DUX4 on HIF1α Pathway in Murine Myoblasts and Myocytes In Vitro

Since many DUX4-mediated signaling alterations are conserved between humans and mice [34], we investigated whether the effect of *DUX4* expression on *HIF1A* mRNA levels observed in human myoblasts is conserved in murine muscle cells. To this aim, we used C2C12-iDUX4 cells derived from mouse C2C12 myoblasts and harboring a DOX-inducible *DUX4* gene [16]. LHCN-M2-iDUX4 and C2C12-iDUX4 at a standard PO_2_ of 21% were both induced using increasing DOX doses (Figure 3A–C). In both cell lines, *DUX4* expression decreased the proportion of HIF1α^+^ nuclei. However, the basal percentage of Hif1α^+^ nuclei in mouse myoblasts (72.9% ± 4.3%) was higher than that in human myoblasts (43.9% ± 2.73%) (Figure 3C). Moreover, in human myoblasts, DUX4 induction by low DOX doses was sufficient to decrease the percentage of Hif1α^+^ nuclei. Indeed, 15.6 ng/mL of DOX resulted in a significant decrease in the percentage of Hif1α^+^ nuclei (31.5% vs. 43.9% in non-induced controls), whereas in mouse myoblasts, the reduction in Hif1α^+^ nuclei could only be seen with 31.3 ng/mL of DOX (Figure 3C). As expected, in mouse myoblasts and myocytes, 62.5 ng/mL of DOX induced the mRNA level of the DUX4 footprint gene *Wfdc3* (Figure 3D,I). In myoblasts, though the change in *Hif1a* expression did not reach statistical significance, we observed decreased levels of *Hif1a* target genes *Vegfa* and *Pdk1* in *DUX4*-expressing cells (Figure 3E)*,* consistent with the reduced percentage of Hif1α^+^ nuclei presented in Figure 3C. In murine myocytes, DUX4 induction significantly decreased the percentage of Hif1α^+^ nuclei (Figure 3H). However, as observed in the human cell model (Figure 2E), no significant changes were observed in *Hif1a*, *Vegfa* and *Pdk1* mRNA levels (Figure 3J). 

### 2.4. Characterization of HIF1α Pathway Modifications upon DUX4 Expression in Adult Mouse Muscle In Vivo

To investigate the potential link between DUX4 and the HIF1α pathway in mature muscle in vivo, we used the DUX4 IMEP mouse model that we had previously developed [45]. In this model, a *DUX4* expression plasmid (*pCIneo-DUX4*) is injected into the mouse *Tibialis Anterior* (TA) hindlimb muscle followed by electroporation (IMEP), leading to local *DUX4* expression and myopathy. First, a dose–response analysis was performed with increasing amounts of *pCIneo-DUX4*, using the backbone control plasmid (*pCIneo*) or a saline solution as negative controls. TA muscles were harvested 7, 14 and 21 days post IMEP and frozen (Appendix A). Cryosections were stained using Hematoxylin–Eosin–Heindehain blue (HEB). The quantification of muscle damage characterized at day 7 by extracellular matrix expansion (fibrosis) and atrophic myofibers (Appendix A) was reported for the total muscle section area and performed as described in [45]. At 14 and 21 days after *pCIneo-DUX4* electroporation, the TA muscle of DUX4 IMEP mice no longer exhibited these histological features but presented many fibers with centrally located nuclei, suggesting muscle regeneration (Appendix A). Upon quantification of the damaged area, we found no statistical difference between the saline and the *pCIneo* control plasmid groups; therefore, we pooled data from both groups into a single control group (Appendix A, right panel). The lowest *pCIneo-DUX4* dose causing a significant increase in the damaged area (median of 20.7%) compared to the control group (median of 6.8%) was 5 µg (*p* < 0.05, ANOVA on ranks followed by Dunn’s post hoc test; Appendix A). To check DUX4 biological activity, we quantified the mRNA level of its mouse target gene, *Wfdc3*. We found no statistical difference in *Wfdc3* mRNA levels between the saline and the *pCIneo* control plasmid groups; therefore, we pooled data from both groups into a single control group. A significant increase in *Wfdc3* mRNA levels was detected by RT-qPCR at days 1, 3, 7 and 14 post injection, confirming *DUX4* expression in the injected TA (Appendix A). However, we could no longer detect an increase in *Wfdc3* mRNA levels at 21 days post injection. We then investigated the Hif1α pathway in this model at 1, 3, 7 and 14 days post injection, with the lowest dose of *pCIneo-DUX4* causing a significant increase in the damaged TA muscle area (5 µg). No significant difference was detected in mRNA levels of *Hif1a* and its target gene *Pdk1* at any timepoint. However, at 1 day post injection only, a significant increase in *Vegfa* mRNA levels was observed in the control mice injected with *pCIneo* as compared to saline. This increase was not detected with *pCIneo-DUX4* injection (Appendix A).

In contrast to the data that we obtained in human myotubes, *DUX4* expression did not affect the Hif1α pathway in mouse adult myofibers in the DUX4 IMEP model at the investigated times post injection and by using a 5 µg dose of *DUX4* expression plasmid. The first hypothesis that explains those divergent results is that DUX4 could have influenced the HIF1α pathway in human muscle cells but not in murine muscle cells. However, we showed that DUX4 could decrease the number of HIF1α^+^ nuclei in murine as well as in human immortalized myoblasts in vitro (Figure 3). We therefore investigated whether HIF1α dysregulation could constitute an early event following *DUX4* expression. To this aim, and to respect the ethical principle of reduction in animal experimentation, we first selected the most relevant acute timepoints in a model in vitro. Unmodified C2C12 murine myoblasts were transfected with *pCIneo-DUX4* because this method was closer to the conditions used in the DUX4 IMEP model in vivo as compared to DUX4-inducible cell models (Figure 4A). To evaluate the kinetics of DUX4 target gene transcription following the transfection, we quantified by RT-qPCR the mRNA levels of two DUX4 target genes, *Wfdc3* and *Zscan4c* (Figure 4B). We could detect a significant increase in *Zscan4c* expression at 5 h and 6 h post transfection of 4- and 17-fold, respectively. There was also an 8-fold increase in *Wfdc3* mRNA levels at 6 h only. The Hif1α pathway was therefore investigated at these timepoints in the DUX4 IMEP model. To increase the model sensitivity, we also increased the dose of *pCIneo-DUX4* up to 20 µg to detect a highly significant (compared to 5 µg) increase in the muscle lesion area (median of 24.7%) (*p* < 0.001, ANOVA on ranks followed by Dunn’s post hoc test; Figure 4C,D and Appendix A). At 6 h, 1 day and 7 days post injection, a significant increase in *Wfdc3* mRNA was detected, confirming *DUX4* expression in the injected TA (Figure 4E). Concerning *Hif1a* mRNA levels, an increase was observed in TA 6 h after injection of *pCIneo-DUX4* and *pCIneo*. At 1 day post injection, *Hif1a* expression was only increased in the *pCIneo-DUX4* group, as compared to the control groups injected with the *pCIneo* plasmid or the saline solution. However, the expression of target genes *Pdk1* and *Vegfa* was not significantly modified by the experimental group or time point (Figure 4F). 

### 2.5. Effect of a Targeted Hif1a Knockdown on DUX4-Mediated Muscle Lesions In Vivo

Since DUX4 deregulated *Hif1a* mRNA levels in mouse TA muscle, we studied the involvement of Hif1α in DUX4-mediated muscle damage in vivo through loss-of-function experiments by using *siRNAs* targeting *Hif1a* transcripts (*siHIF1α*) in the DUX4 IMEP model. We first checked s*iHIF1α* efficiency through a dose–response analysis (Appendix A) 1 day post injection, when morphological alterations were not yet observable (as we described in [45]). We found that injection of 2 µg of *siHIF1α* allowed significant two-fold downregulation of *Hif1a* mRNA levels as compared to the TA injected with the control *siRNA* (*siCTL)* or saline (Figure 5A and Appendix A). To evaluate the implication of Hif1α in muscle lesions induced by DUX4, we first used the global quantification of the damaged surface area in the TA muscle of DUX4 IMEP mice at 7 days as an outcome measure, namely when a local muscle lesion was observable. At this time point, the lesion was characterized by extra-cellular matrix (ECM) expansion (fibrosis) and a high number of atrophic fibers [45]. Here, the *pCIneo-DUX4* DNA was injected alone or in combination with 2 µg of *siCTL* or *siHIF1α* (Figure 5B–H). A significant 9.4% increase in muscle lesion area was observed in the *siHIF1α* group, as compared to the s*iCTL* and saline groups (Figure 5B,C). The muscle lesion in the *siHIF1α* mouse group was characterized by exacerbated ECM expansion (as shown by HEB staining, Figure 5B,C). However, concerning the diameter of remaining myofibers, TA cross-sectional area (CSA) and fiber size distribution were similar in the *siHIF1α* and the *siCTL* groups and were not significantly different from the control DUX4 IMEP mice having not received any *siRNA* (Figure 5D–F). We also evaluated the kinetics of DUX4 target gene transcription following the IMEP procedure. Within the damaged TA muscle at 7 days, we did not detect any significant difference in *Wfdc3* mRNA levels between all groups (Figure 5G). Similarly, we did not observe any difference for the mRNAs of *Hif1a* or its target genes *Pdk1* and *Vegfa* among the groups (Figure 5H).

## 3. Discussion

### 3.1. DUX4-Induced HIF1α Pathway Disturbances Depend on the Differentiation State of Human Muscle Cells

The hypoxic response pathway has been described as critically disturbed in FSHD muscles [15,34]. HIF1α, the key driver of this response, was presented as one of the main actors of DUX4-induced myoblast death [39]. In the present study, we confirmed that HIF1α nuclear protein levels were altered in human muscle cells upon *DUX4* expression. However, these alterations differed according to the stage of myogenic differentiation. Indeed, in proliferating myoblasts, HIF1α was downregulated at the mRNA and protein levels upon *DUX4* expression, but in contrast, DUX4 induced HIF1α and its pathway in myotubes. At the intermediate differentiation stage (myocytes), HIF1α expression was not significantly modified by *DUX4* expression. Previous data have suggested that DUX4-mediated cell death requires induction of the HIF1α pathway in another model of DUX4-inducible human myoblasts (iDUX4-MB135). Indeed, Lek et al. showed HIF1α protein stabilization and nuclear localization upon *DUX4* expression as well as colocalization of HIF1α in nuclei with high DUX4 immunolabeling, suggesting that a threshold of *DUX4* expression is necessary to trigger HIF1α stabilization [39]. In our study, this positive relationship was observed in DUX4-expressing human LHCN-M2-iDUX4 myotubes but not in proliferating myoblasts. The use of different muscle cell lines is not expected to strongly influence the DUX4–HIF1α axis, contrary to culture conditions. Indeed, the addition of dexamethasone to the culture medium (as per Lek et al.) was found to lower *DUX4* expression in FSHD myoblasts [47]. Similarly, dexamethasone treatment of human myoblasts for 24 h decreased the expression of *VEGF*, a HIF1α target gene [48]. In contrast, in the present study, we cultured myoblasts in a medium without dexamethasone, based on the original publication that described the inducible LHCN-M2-iDUX4 cell line [17]. 

The different regulation of HIF1α upon *DUX4* expression in myoblasts and myotubes can partly be explained by distinct basal levels of HIF1α according to the differentiation stages. Indeed, basal Hif1α protein levels were shown to be higher in murine myoblasts compared to myotubes. This is unlikely due to different gene expression levels since no change in mRNA abundance was seen [49]. We recently confirmed that the percentage of HIF1α^+^ nuclei decreased during human myoblast differentiation [40]. In the same study, we showed that DUX4 suppressed HIF1α-mediated precocious differentiation of human myoblasts. Several studies of transcription profiles or myocyte fusion index have shown that myogenesis impairment and defects in regeneration are typical features of FSHD muscle biopsies or cell cultures [1,15,17,20,35,43]. Particularly, Bosnakovski et al. [17] characterized the LHCN-M2-iDUX4 model and showed that low levels of DUX4 negatively impacted the myogenic program and myotube differentiation. Our data suggest that DUX4-mediated disturbances of the HIF1α pathway in myoblasts and myotubes can contribute to these pathological characteristics of FSHD muscle cells. In keeping with the idea that DUX4 mediates HIF1α inhibition in proliferating myoblasts, we observed a particularly low percentage of DUX4^+^/HIF1α ^+^ nuclei at this stage. In myotubes, this percentage increased but only reached half of HIF1α^+^ nuclei co-expressing DUX4. Direct or indirect mechanistic hypotheses might be suggested to explain this discrepancy: (I) In the case of direct DUX4-mediated transcription activation of the *HIF1A* gene, we expect different kinetics between DUX4 protein synthesis in the cytoplasm, its nuclear translocation, activation of *HIF1A* gene expression, HIF1α protein synthesis and nuclear translocation (as we described for other DUX4 target genes in [50]). Moreover, the kinetics of nuclear/cytoplasm shuttling or protein turnover could also differ. Therefore, the expression dynamics and asynchronous regulation of DUX4 and HIF1α nuclear location and half-life could explain why both proteins were detected either individually in separate nuclei or together in identical nuclei [50]. Indirect mechanisms may also be suggested: (II) DUX4 could activate the HIF1α pathway in satellite cells by disturbing the ability of PAX7 to regulate its transcriptional network [15]. (III) The HIF1α protein could also be stabilized by ROS through the inhibition of prolyl hydroxylase enzyme (PHDs) activities [51]. Indeed, several clinical and experimental studies have indicated both systemic and muscle-specific oxidative stress in FSHD [18,20,25,52,53,54,55]. A recent study also showed high mitochondrial ROS in FSHD myogenic cells [18]. DUX4 could either favor ROS generation or disrupt anti-oxidant processes leading to a global ROS increase that could stabilize HIF1α proteins even in myotube nuclei that lack DUX4.

### 3.2. DUX4-Induced PDK1 Disturbances Involved HIF1α-Dependent and -Independent Processes

We further investigated the effect of DUX4 on the HIF1α pathway through an analysis of its target gene expression. *VEGFA* and *PDK1* were globally repressed in proliferating human myoblasts, unchanged in myocytes and induced in myotubes expressing DUX4. These results are in accordance with variations in HIF1α mRNA and protein levels at the same stages of the myogenic differentiation and suggest a regulatory switch during the differentiation process. The turning point seems to occur at the myocyte stage, namely around the first two days of differentiation in the culture. This transition could involve HIF1α-dependent metabolic changes. Indeed, HIF1α is known to upregulate genes encoding enzymes that promote a glycolytic metabolism [56,57,58]. It is well established that most stem cell types reside in hypoxic niches where HIF1α controls pluripotency gene expression, promotes glycolytic metabolism and inhibits mitochondrial biogenesis. However, an additional layer of complexity pertaining to HIF1α dysregulation by DUX4 is the fact that myogenic differentiation itself involves metabolic reprogramming, evidenced by gradual metabolic switching from predominantly glycolytic myoblasts toward aerobic myotubes, which mostly rely on oxidative phosphorylation (OXPHOS; reviewed by [59]). Since myogenesis involves an increase in mitochondrial mass [60,61], the known perturbation of mitochondrial function and biogenesis by DUX4 [18,42] could further affect HIF1α indirectly through altering oxygen consumption in the mitochondrial respiratory chain. This may explain the differential effects of DUX4 on HIF1α stabilization in distinct myogenic developmental stages, which are characterized by distinct core metabolic setups. 

DUX4 was found to disturb myogenesis via its target gene activation or repression to orchestrate a transcriptome characteristic of a less differentiated cell state [62]. Therefore, since *DUX4* expression activates embryonic genes and the embryo metabolism is predominantly glycolytic, abnormal activation of glycolysis through HIF1α could contribute to metabolic disturbances in FSHD muscle cells, specifically in myotubes that rely on OXPHOS much more than myoblasts. Accordingly, Heher et al. reported that DUX4-induced changes in oxidative metabolism impaired muscle cells in FSHD and that this phenomenon was amplified when metabolic adaptation to varying O_2_ tension was required [18]. 

On the other hand, HIF1α-independent processes may also be involved in DUX4-mediated PDK1 disturbances at the protein level. Indeed, regardless of the differentiation stage, DUX4 induced a decrease in PDK1 protein abundance. In contrast, in myotubes, *PDK1* mRNA levels were induced by DUX4, suggesting post-translation regulation involving factors independent of HIF1α transcriptional activity. Importantly, PDK1 is a major metabolic regulator of glycolytic versus oxidative metabolism and is mainly located in the mitochondria [63]. Actual evidence for mitochondrial dysfunction was found in FSHD muscle, where impaired energy metabolism was linked to alterations in mitochondrial ultrastructure and subcellular distribution [52]. Furthermore, a dynamic transcriptomic analysis identified that suppression of the PGC1α–ERRα axis, a critical component of the mitochondrial biogenesis pathway, is associated with the myogenesis defect in FSHD [42]. Of note, a recent proteomics study of FSHD muscle cells revealed higher abundance but slower turnover of mitochondrial respiratory complexes and mitochondrial ribosomal proteins, indicating an accumulation of older, less viable mitochondria [64]. Indeed, FSHD patients display impaired muscle oxygenation [65] and have lower resting metabolic rates [66], thus potentially linking metabolic stress and altered tissue oxygen tension with perturbed metabolic switching. This is supported by transcriptomic data from FSHD muscle biopsies, showing enrichment for disturbed mitochondrial pathways, the molecular response to hypoxia and transcriptional repression of the mitochondrial genome [18]. Mechanistically, the alteration of mitochondrial ROS metabolism is correlated with mitochondrial membrane polarization and myotube hypotrophy. DUX4-induced mitochondrial dysfunction occurs before apoptosis through mitochondrial ROS generation and affects hypoxia signaling via complex I [18]. Therefore, mitochondrial dysfunction along with DUX4-mediated mitochondrial ROS production could lead to *PDK1* post-transcriptional deregulation, e.g., protein degradation through the proteasome or autophagy [67].

DUX4-mediated VEGF disturbances are in accordance with corresponding HIF1α expression patterns during muscle cell differentiation. This mediator of angiogenesis is induced by hypoxic and ischemic conditions [68,69] through HIF1α [57,58] and contributes to myoblast differentiation [70] and skeletal muscle regeneration [71,72,73,74]. Even if VEGF was primarily studied here to assess HIF1α transcriptional activity, our results highlight an interest in further investigating the HIF1α–VEGF axis in FSHD, for which evidence of vascular troubles was reported. Notably, patient muscle biopsies exhibit a decrease in capillary density [75]. Moreover, FSHD is characterized by an impaired oxygen demand during exercise, and lower oxygen consumption is related to oxidative stress [54]. Moreover, patients affected with infantile FSHD can present exudative retinopathy due to retinal telangiectasias [76,77,78]. 

### 3.3. The DUX4–Hif1α Axis in Murine Muscle Models In Vitro and In Vivo

Our data indicate that DUX4 inhibits the HIF1 pathway both in human muscle cells and C2C12 murine myoblasts. However, we must consider that characteristics of immortalized C2C12 cells are not strictly identical to satellite cell-derived primary mouse myoblasts [79]. 

We also showed that DUX4 induced the HIF1 pathway in mature muscle in mice, as observed in human myotubes. Even if muscle fibers and myotubes constitute two distinct differentiation steps, this point is of interest because most therapeutic strategies need testing in vivo in preclinical models. Here, we used the DUX4 IMEP mouse model, in which a *DUX4* expression plasmid is injected and electroporated into a TA muscle, inducing dose-dependent local myopathy. In this model, *Hif1a* mRNA levels were increased in both *pCIneo-DUX4* and *pCIneo* IMEP mice 6 h post injection. The increase in the control plasmid group was likely linked to damage induced by the IMEP procedure. Indeed, activation of Hif1α signaling was observed in injured TA muscle, increasing from day 1 to day 7 after cardiotoxin injection [80]. One day post injection, the increased *Hif1a* expression was only maintained in the *pCIneo-DUX4* group, meaning that this increase was DUX4-related. In summary, *DUX4* expression in murine mature muscle induces early and transient *Hif1a* overexpression. However, no modification could be detected in the expression of HIF1α target genes *Pdk1* and *Vegfa*. We cannot rule out that this could result from limitations of the IMEP model (e.g., variability among animals and local transgene expression levels in aggregated data. However, given that the model sensitivity was sufficient to highlight significant changes in *Hif1a* expression, our data suggest that both HIF1α-dependent and -independent factors can influence *Pdk1* and *Vegfa* expression levels. One hypothesis concerning these interfering factors is activation of the muscle regeneration process [73]. Interestingly, Vegf was shown to play a role in skeletal myofiber regeneration in vivo. Concerning *Pdk1*, other transcription factors are known to regulate *its* expression, such as C-Myc or the Wnt pathway [81,82,83,84].

### 3.4. Targeted Hif1α Knockdown in Mice Exacerbated DUX4-Induced Muscle Fibrosis

In the IMEP model, muscle alterations caused by one local boost of *DUX4* expression constitute an easy read-out through semi-automated histological quantification of the damaged area by color thresholding [45]. Moreover, targeting the TA in this local FSHD model is pertinent, as it is one of the most affected leg muscles in patients [85]. Here, to specifically evaluate the involvement of Hif1α in the development of DUX4-induced muscle lesions, in this model, we performed a loss-of-function study with *siRNAs* against *Hif1a* mRNA (*siHIF1α*). The amount of *siRNA* we used caused about a two-fold *Hif1a* mRNA reduction. However, *Hif1a* knockdown did not reduce the proportion of atrophic fibers in the DUX4 IMEP mice. By contrast, we observed a significant increase in the muscle lesion area characterized by exacerbated extra-cellular matrix expansion. The extension of the muscle lesion with *Hif1a* knockdown can be explained by (I) a synergistic and negative effect of DUX4 and *siHIF1α* on the myofiber itself or (II) a loss of the critical role of Hif1α in muscle regeneration. Indeed, the role of Hif1α in the regeneration process was first described in murine myoblasts [86] and in *Hif1a* KO mice [87,88]. Moreover, we recently showed that HIF1α is necessary for the early myogenic differentiation of human myoblasts [40]. Since FSHD was recently considered a satellite cell-opathy [89,90], further studies should investigate the impact of *HIF1A* knockdown, notably in satellite cellsand with precise kinetics following the local boost of *DUX4* expression in vivo.

The exacerbation of the muscle lesion observed here upon *Hif1a* knockdown contrasts the results of Lek et al. [39] obtained with FDA-approved HIF-signaling inhibitors. In these studies, FSHD-like zebrafish models (either single-cell embryos injected with *DUX4* mRNA or transgenic fish eggs with inducible *DUX4* expression) were used, and an improvement of muscle structure and function was observed. However, most of the drugs used were indirect inhibitors of HIF1α and had multiple effects, notably on protein turnover, and only short-term effects were investigated. For chronic treatment of muscle disorders, the negative effect of indirect HIF1α inhibitors on protein synthesis should be avoided given the risk of exacerbating muscle atrophy. In addition, as we mentioned in [38], drugs developed to interfere with HIF1α expression or activity are developed to induce targeted cancer cell death. This highlights the importance of the context of use when considering the action on the HIF1α pathway. Although DUX4 was induced at an early embryonic stage in the zebrafish models, our results were obtained by inducing *DUX4* expression directly in mature muscle fibers in our mouse model. Both DUX4 and HIF1α are known to have specific activities during these two different development stages. This could explain the discrepancies observed in the results obtained. Because of its critical role in many skeletal muscle mechanisms, HIF1α remains challenging to target from a therapeutic perspective. 

In conclusion, our study sets the basis for further investigations on the role of HIF1α in relationship with DUX4 in FSHD (Figure 6). We found that this link differs according to the muscle cell differentiation stage. Our results also suggest that this axis also exists in mouse muscle cell cultures and adult muscle. Finally, we found that *Hif1a* knockdown in an FSHD mouse model unexpectedly expanded the DUX4-mediated muscle damaged area, particularly through exacerbation of fibrosis in the lesion sites. This indicates that the DUX4–HIF1α axis is not as simple as expected and that targeting HIF1α might be challenging in the context of FSHD therapeutic approaches. Finally, given (I) the regeneration defects in FSHD [89,90,91], (II) the role of HIF1α in this process [38,40] and (III) the impact of *DUX4* expression on the HIF1α pathway depending on the differentiation state, further investigations, especially in cellular actors of muscle regeneration (e.g., satellite cells) appear critical for a better understanding of FSHD-associated muscle regeneration disturbances. Moreover, regarding the pivotal role of HIF1α in muscle metabolism, it will be important to clarify whether metabolic disturbances could contribute to the development of muscle dysfunction in FSHD. Such studies would provide more in-depth mechanistic insights into the FSHD pathogenic network and could suggest additional therapeutic targets.

## 4. Materials and Methods

### 4.1. Cell Culture

Immortalized human myoblast cell lines (LHCN-M2 and LHCN-M2-iDUX4 [17]) were kindly provided by Prof. M.Kyba (Lillehei Heart Institute, University of Minnesota, MI, USA). Cells were cultured in a DMEM F12 (BioWest, Nuaillé, France) proliferation medium supplemented with 20% FBS (Biowest) and 1% Penicillin/Streptomycin (P/S, Thermo Fisher Scientific, Waltham, MA, USA) at 37 °C in a 5% CO_2_ and atmospheric O_2_ levels (standard PO_2_ of 21%). For myogenic differentiation, cells were cultured on Matrigel-coated dishes (Corning, Corning, NY, USA) in a proliferation medium until 100% confluence. Cells were then washed once with PBS and differentiated for two days for myocytes and four days for myotubes using a differentiating medium (DMEM F12) (Corning), supplemented with human insulin at 10 µg/mL (Sigma-Aldrich, St. Louis, MO, USA), bovine apo-transferrin at 100 µg/mL (Sigma-Aldrich, Merck KGaA, Darmstadt, Germany) and 1% Penicillin/Streptomycin (P/S, Thermo Fisher Scientific).

Immortalized mouse myoblast cell lines (C2C12 and C2C12-iDUX4 [16]) were kindly provided by Prof. M. Kyba. They were cultured in a DMEM high-glucose (BioWest) proliferation medium supplemented with 10% FBS (Biowest) and 1% Penicillin/Streptomycin (P/S, Thermo Fisher Scientific) at 37 °C in a 5% CO_2_ atmosphere. For differentiation into myocytes, cells were cultured on Matrigel-coated dishes (Corning) in a proliferation medium until 100% confluence. Cells were then washed once with PBS and cultured for two days in a differentiating medium (DMEM high-glucose) (Biowest), supplemented with human insulin at 10 µg/mL (Sigma-Aldrich), bovine apo-transferrin at 100 µg/mL (Sigma-Aldrich) and 1% Penicillin/Streptomycin (P/S, Thermo Fisher Scientific). Myogenin was immunolabelled as a marker of early differentiation (Appendix A).

### 4.2. Viability Test

For the Vybrant^®^ MTT Cell Proliferation Assay Kit (Thermo Fisher Scientific), LHCN-M2 iDUX4 and LHCN-M2 cells were seeded in a 96-well plate and induced for 24 h with doxycycline (DOX). Cells were then incubated for 2 h with MTT 1.2 mM reagent diluted in a proliferation medium at 37 °C. After that step, the medium was replaced by DMSO to solubilize the formazan product, and the plate was incubated for 10 min at 37 °C under agitation. The absorbance was then measured by a spectrophotometer (VERSA max-SoftMax Pro, Molecular Devices, San José, CA, USA) at 540 nm.

For Cell Counting Kit-8 (Sigma-Aldrich), LHCN-M2 iDUX4 and LHCN-M2 myoblasts were seeded in a 96-well plate and induced for 24 h with DOX. Cells were then incubated for 1 h with a CCK-8 solution diluted in a proliferation medium at 37 °C. Absorbance was then measured by a spectrophotometer (VERSA max-SoftMax Pro) at 450 nm.

### 4.3. Myoblast Transfection

A total of 10^5^ C2C12 mouse cells were seeded in 6-well plates and transfected 24 h later in Opti-MEM (Invitrogen|Thermo Fisher Scientific, Waltham, MA, USA) with 5 µL of Lipofectamin 2000 (Invitrogen) and 1600 ng of DNA vector according to the manufacturer’s instructions.

### 4.4. Immunofluorescence

Cells previously seeded in 6-well plates on glass slides were fixed with 4% paraformaldehyde/PBS for 10 min, permeabilized with 0.5% TritonX-100/PBS for 10 min and incubated with blocking solution (5% normal goat serum (Biowest), TritonX-100/PBS) for 1 h at room temperature. Cells were then incubated with primary antibodies (anti-DUX4 9A12 MABD116; 1:100, Merck Millipore, MA, USA ([50]); HIF1α ab179483, 1:500, Abcam, Cambridge, UK) at 4 °C overnight. They were subsequently rinsed in PBS and incubated with secondary antibodies Alexa 555 Goat anti-rabbit IgG (1:500, Biotium, Fremont, CA, USA) and Alexa 488 Goat anti-mouse IgG (1:500, Biotium) at room temperature for 1 h. Immunolabeled cells were rinsed in PBS and mounted with EverBrite Mounting Medium with DAPI (Biotium) for nuclear staining. Pictures were taken with a Nikon Eclipse 80i microscope (Nikon Corporation, Tokyo, Japan) and merged using NIS-Elements software (https://www.microscope.healthcare.nikon.com/fr_EU/products/software/nis-elements).

### 4.5. qPCR

RNA was extracted using Trizol reagent (Invitrogen) according to the manufacturer’s directions. The total RNA was then treated with DNAse I (amplification grade, Thermo Fisher Scientific). cDNAs were synthetized using the Maxima First Strand cDNA Synthesis Kit (Thermo Fisher Scientific). All qPCRs were performed in triplicate using SYBR Green FastStart Essential DNA Green Master (Roche, Bâle, Swiss) and corresponding primers (Eurogentec, Seraing, Belgium) (see Appendix A). Cycling conditions were as follows: initial denaturation step at 95 °C for 10 min, followed by 40 cycles of 15 s at 95 °C and 60 s at primer Tm. qPCR results were analyzed with LightCycler 96 software (Roche). Quantifications were performed using the 2^−∆∆Ct^ method.

### 4.6. Western Blot

Cells were lysed using RIPA buffer. Proteins were separated on 12% SDS-PAGE gels for 3 h at 100 V and transferred to a nitrocellulose membrane for 1 h 45 at 260 mA. Membrane blocking was performed using 5% nonfat dry milk diluted in TBST-T. Primary (PDK1, ab110025, Abcam) and secondary HRP-conjugated antibodies (NA931, Amersham ECL, VWR International, Radnor, PA, USA) were diluted in 5% skim milk in TBST and incubated overnight at 4 °C and for 1 h at room temperature, respectively. The HRP signal was visualized using Supersignal West Femto Max. Sensitivity Kit (Thermo Fisher Scientific) and the Fusion FX7 spectra (Vilber, France). Densitometry was performed using ImageJ software (https://imagej.net/ij/). The densitometry signal was normalized to the total proteins stained by Ponceau red.

### 4.7. ELISA

Human VEGF concentrations were measured with the ELISA kit DVE00 (R&D Systems, Minneapolis, MN, USA) in LHCN-M2-iDUX4 culture media according to the manufacturer’s instructions.

### 4.8. Ethics Statement

All animal experiments met the Belgian national standard requirements regarding animal care and were conducted in accordance with the Ethics and Welfare Committee of the University of Mons (reference number LE018/02).

### 4.9. IMEP Mouse Model

Female C57BL/6 mice, aged between 8 and 12 weeks, were purchase from Charles River laboratories (France). Mice were housed in a conventional animal colony and maintained at 35–40% relative humidity with a constant room temperature (21 °C) and a natural day/night light cycle (12 h/12 h). Food and water were provided ad libitum, and animals were subjected to an adaptation period of 7 days before experiments. The mouse IMEP model was generated as we previously developed in [45] on the basis of the naked DNA electroporation procedure described in [92]. Briefly, tibialis anterior (TA) muscles were injected with 40 µg of hyaluronidase. Each TA was then injected with either naked plasmid DNA alone or complemented with *siRNAs* targeting *Hif1a* mRNA (*siHIF1α*, FlexiTube #1027416, Qiagen, Hilden, Germany) or the control (*siCTL*; Qiagen, #1027280, “all star negative control”). As in [40], a mix of 4 *siRNAs* directed against the *HIF1a* mRNA was used. Redundancy experiments using several distinct *siRNAs* targeting different sequences of the same mRNA prevented sequence-derived off-target effects. *siRNA* were electroporated using an EMKA stimulator. Mice were checked daily and then sacrificed by an intraperitoneal injection of Nembutal (Kela, Belgium).

### 4.10. Tissue Preparation and Histology

At the indicated euthanasia time points, right and left TAs were removed, embedded in OCT compound (VWR International) and frozen in liquid-nitrogen-cooled isopentane. Cryostat sections that were 8 µm thick from the proximal and medial TA were cut using a Leica cryotome, and sections were stained with Hematoxylin–Eosin–Heidenhain blue (HEB) to evaluate the percentage of muscle lesions. HEB staining consisted of basic Hematoxylin–Eosin coloration followed by 45 s of incubation in Heidenhain’s Blue staining (mix of orange G and Aniline Blue, Sigma-Aldrich), allowing intense blue labeling of fibrotic fibers and collagenous tissues. Slides were then scanned using the NanoZoomer-SQ Digital slide scanner (Hamamatsu Photonics, Japan). Images were processed as described in [45]. Transgene expression and lesion distribution were characterized in [45]. Each myofiber cross-section area (CSA) was measured on the whole muscle section by using ImageJ software (https://imagej.net/ij/). The total CSA was calculated. Myofibers were also classified into clusters according to their CSA. Myofiber CSA distribution was presented along with the cumulative percentage of myofibers in each cluster.

### 4.11. Statistical Analysis

Normality tests (Shapiro–Wilk) were performed on each data set to assess the data distribution, and thus, appropriate statistical tests could be chosen. Differences were considered statistically significant at a *p*-value < 0.05. All data were represented as the mean ± SEM or as boxplots (5th and 95th percentile) for parametric or non-parametric statistical tests, respectively. Statistical analyses were performed using GraphPad Prism software, version 8.02 and SigmaPlot software, version 14.

## Figures and Tables

**Figure 1 ijms-25-03327-f001:**
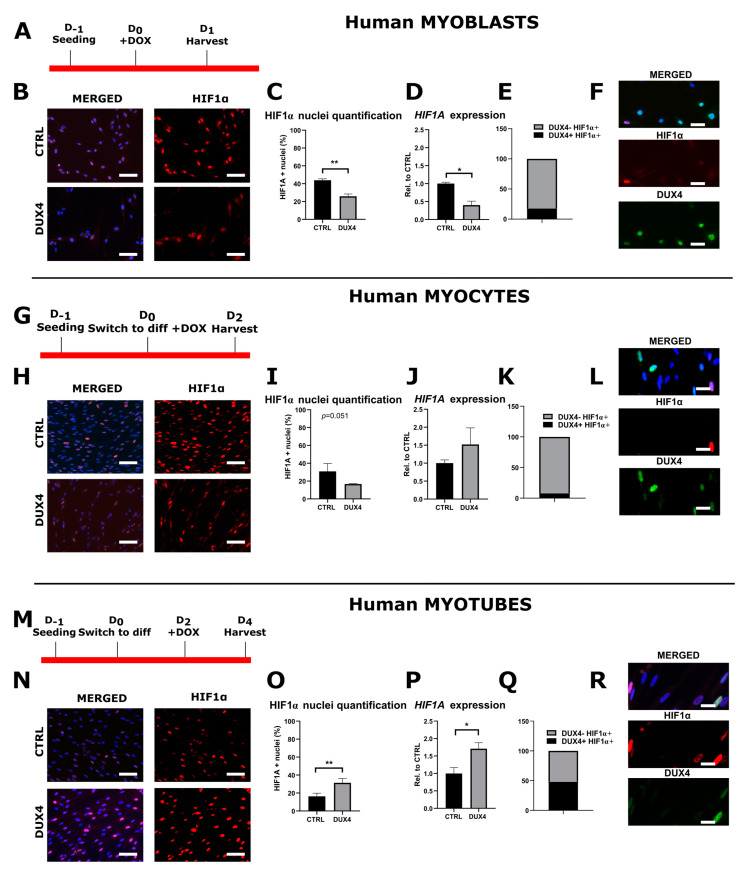
The differential effect of DUX4 on HIF1α expression and protein level in human LHCN-M2-iDUX4 muscle cells depends on the stage of differentiation. LHCN-M2-iDUX4 myoblasts were cultured and seeded as described in [40] at a standard PO_2_ of 21%. *DUX4* expression was induced by the addition of 62.5 ng/mL of doxycycline (DOX) to the culture medium. For differentiation, cells at confluence were switched to a differentiation medium for two (myocytes) or four days (myotubes). Cells were fixed in 4% paraformaldehyde (PAF), and immunofluorescence (IF) was performed with antibodies directed against HIF1α or DUX4 and appropriate secondary antibodies coupled to Alexa Fluors. (**A**,**G**,**M**) Experiment time courses. (**B**,**H**,**N**) Representative fields showing HIF1α-positive (HIF1α^+^) nuclei (red IF). DAPI was used to visualize nuclei (blue). Scale bar = 100 µm. (**C**,**I**,**O**) Quantification of HIF1α^+^ nuclei normalized to the total number of nuclei (DAPI staining). Mean ± SEM, ** *p* < 0.01, *t*-test. (**D**,**J**,**P**) Relative *HIF1A* mRNA level normalized to *RPLP0.* Mean ± SEM, * *p* < 0.05, *t*-test. N = 4 for myoblasts, N = 3 for myocytes and myotubes. (**E**,**K**,**Q**) Proportion of DUX4^+^ nuclei among HIF1α^+^ nuclei. (**F**,**L**,**R**) Representative field showing HIF1α^+^ (red IF) and DUX4^+^ (green IF). Nuclei were stained with DAPI (blue). Scale bar = 50 µm. All experiments were performed on 3 independent cultures, each at least in triplicate. The total numbers of counted nuclei were, on average, 4719 for myoblasts, 5689 for myocytes and 24,486 for myotubes.

**Figure 2 ijms-25-03327-f002:**
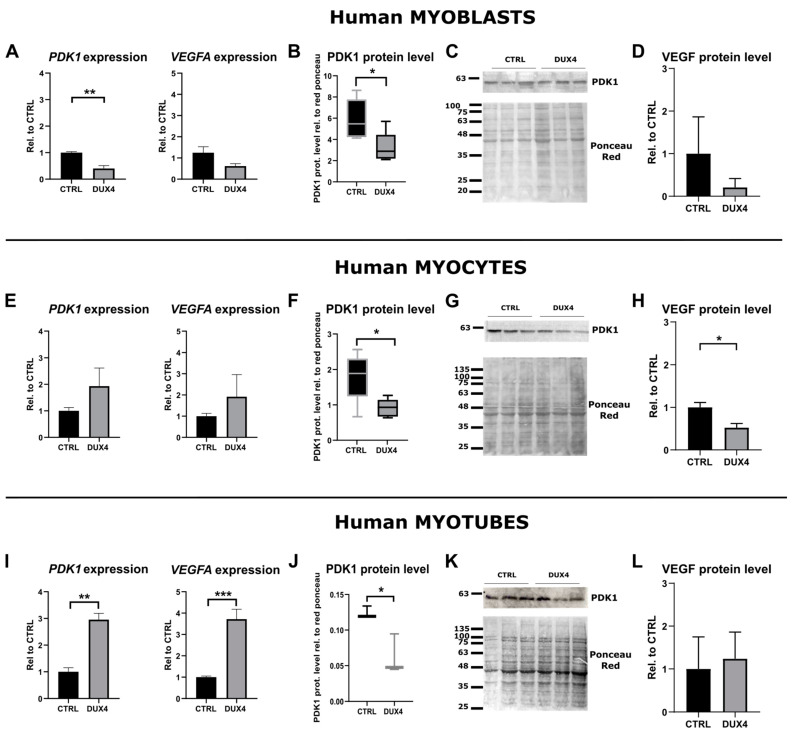
Effect of DUX4 induction on HIF1α target genes in human LHCN-M2-iDUX4 muscle cells. Cell culturing, induction of *DUX4* expression by doxycycline and myogenic differentiation were performed at a standard PO_2_ of 21%, as in Figure 1. (**A**,**E**,**I**) Expression levels of *PDK1* and *VEGFA* mRNAs. Quantifications were performed by RT-qPCR and normalized to *RPLP0*. Mean ± SEM, ** *p* < 0.01, *** *p* < 0.001, *t*-test. N = 4 for myoblasts, N = 3 for myocytes and myotubes. (**B**,**F**,**J**) PDK1 protein level determined by Western blot. Densitometry signal was normalized to total protein stained by Ponceau red. * *p* < 0.05, rank sum test, N = 6 for myoblasts and myocytes, N = 3 for myotubes. (**C**,**G**,**K**) Representative Western blot and Ponceau red staining for PDK1 detection. (**D**,**H**,**L**) VEGF protein level determined by ELISA on the culture medium. Mean ± SEM, * *p* < 0.05, *t*-test. N = 3.

**Figure 3 ijms-25-03327-f003:**
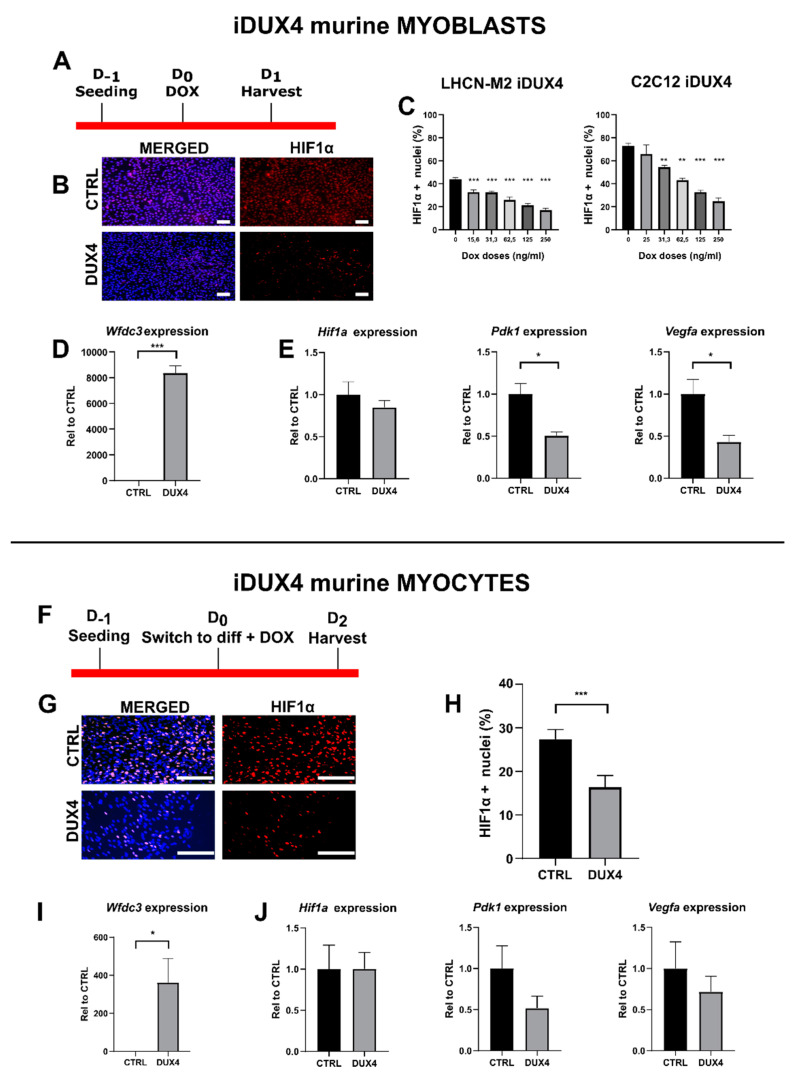
Effect of DUX4 on the Hif1α pathway in murine myoblasts and myocytes. (**A**–**E**) C2C12-iDUX4 murine myoblasts. A total of 25,000 or 200,000 cells were seeded per well in 24-well or 6-well plates, respectively, and grown at a standard PO_2_ of 21%. After 24 h, *DUX4* expression was induced for 24 h with increasing doses of doxycycline (DOX, ng/mL). HIF1α was detected by immunofluorescence (IF). For comparison, LHCN-M2-iDUX4 myoblasts were cultured as in Figure 1, and *DUX4* expression was induced for 24 h with increasing doses of DOX. (**F**–**J**) C2C12-iDUX4 murine myocytes. A total of 750,000 cells were seeded per well in 6-well plates. After 24 h, cells were switched to the differentiation medium for two days. *DUX4* expression was induced with 62.5 ng/mL of DOX for 48 h. (**A**,**F**) Experiment time courses. (**B**,**G**) Representative fields showing Hif1α^+^ nuclei (red IF). Nuclei were stained with DAPI (blue). Scale bar = 100 µm. (**C**) Quantification of Hif1α^+^ nuclei normalized to the total number of nuclei (DAPI staining) in LHCN-M2-iDUX4 myoblasts and C2C12-iDUX4 myoblasts. Mean ± SEM, ** *p* < 0.01, *** *p* < 0.01, one-way ANOVA with Holm–Sidak post hoc test vs. the control (DOX: 0 ng/mL). (**H**) Quantification of HIF1α^+^ nuclei normalized to the total number of nuclei (DAPI staining) in C2C12- iDUX4 myocytes. Mean ± SEM, *** *p* < 0.001, *t*-test. (**D**,**I**) Expression level of *Wfdc3* mRNA. Quantifications were performed by RT-qPCR and normalized to *Rplp0*. Mean ± SEM, * *p* < 0.05, *** *p* < 0.001, *t*-test. (**E**,**J**) Expression levels of *Hif1a*, *Pdk1* and *Vegfa* mRNAs. Quantifications were performed by RT-qPCR and normalized to *Rplp0*. Mean ± SEM, * *p* < 0.05, *t*-test. Experiments were performed on 3 independent cultures, each in triplicate (N = 3). The total numbers of counted cells were on average 6809 for myoblasts and 3095 for myocytes.

**Figure 4 ijms-25-03327-f004:**
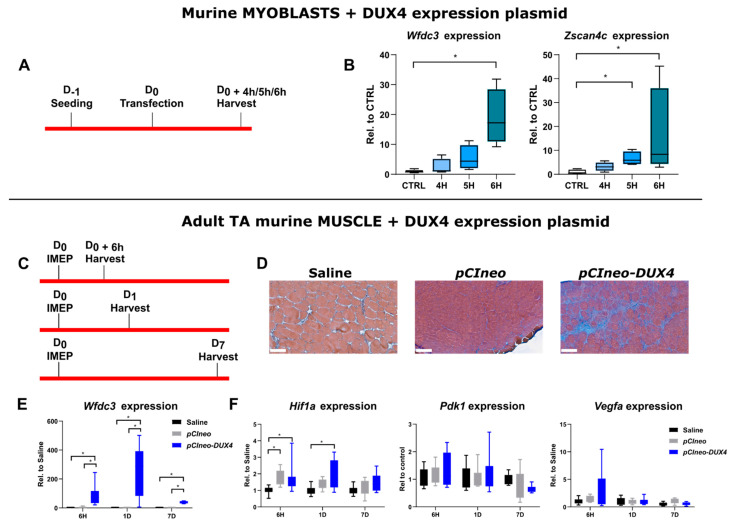
Early effects of *DUX4* expression on HIF1α pathway in vivo in the DUX4 IMEP (Intramuscular Electroporation) mouse model with a high dose of *DUX4* expression. (**A**,**C**) Experiment time courses. (**B**) mRNAs of DUX4 target genes *Wfdc3* and *Zscan4c* were quantified by RT-qPCR in C2C12 myoblasts 4, 5 and 6 h post transfection with *pCIneo-DUX4*. Quantifications were normalized to *Rplp0*. N = 5 for control group, and N = 4 for 4 h, 5 h and 6 h groups. Results are presented as boxplots. * *p* < 0.05. Kruskal–Wallis followed by Dunn’s post hoc test. (**D**) Representative sections of TA electroporated with saline solution (Left), 20 µg of *pCIneo* (Middle) or *pCIneo-DUX4* plasmid (Right) stained using Hematoxylin–Eosin–Heindehain blue (HEB). Scale bar = 100 µm. (**E**) Effect of DUX4 induction on the level of *Wfdc3* mRNA in the IMEP model. mRNA levels were quantified by RT-qPCR and normalized to *Rplp0*. Results are presented as boxplots. * *p* < 0.05. Kruskal–Wallis followed by Dunn’s post hoc test. N = 8 for each group. (**F**) Effect of DUX4 induction on the level of Hif1α pathway mRNAs *Hif1a*, *Pdk1* and *Vegfa* in the IMEP model. mRNA levels were quantified by RT-qPCR and normalized to *Rplp0*. Results are presented as boxplots. * *p* < 0.05. Kruskal–Wallis followed by Dunn’s post hoc test. N = 8 for each group.

**Figure 5 ijms-25-03327-f005:**
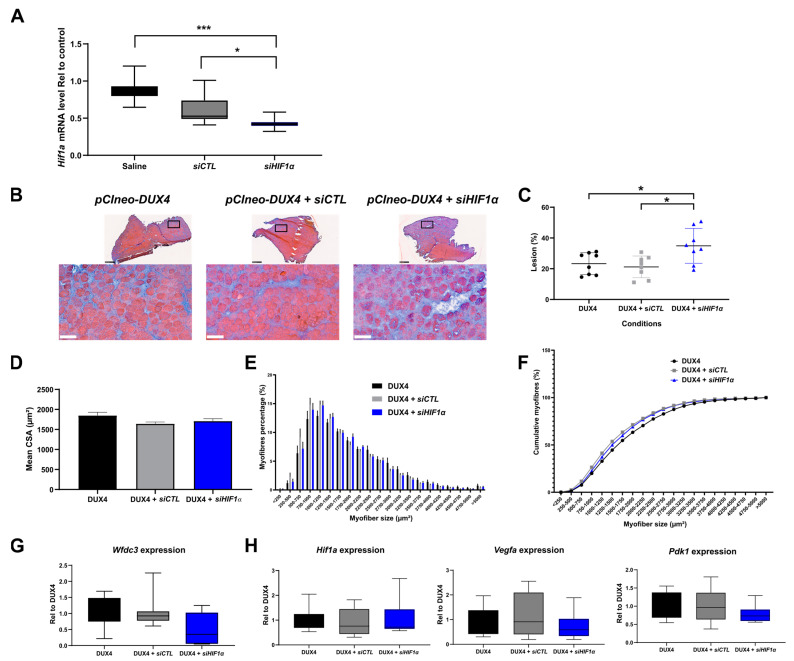
Involvement of Hif1α pathway in DUX4-induced muscle damage. (**A**) Efficiency of *siRNA* directed against *Hif1a* mRNA (s*iHIF1α*). The TA muscle was electroporated with either saline solution, *siCTL* or *siHIF1α*. TA muscles were harvested 1 day after the IMEP procedure. *Hif1a* mRNA level was quantified by RT-qPCR and normalized to *Rplp0*. Mean ± SEM, * *p* < 0.05, *** *p* < 0.001. Kruskal–Wallis followed by Dunn’s post hoc test. N = 10 per group. (**B**) Representative cryosections of TA electroporated with 20 µg of *pCIneo-DUX4* plasmid in combination or not with 2 µg of *siCTL* or s*iHIF1α*. TA muscles were harvested 7 days after the IMEP procedure. Muscle sections were stained with HEB. Top: global view of the muscle sections; scale bar = 500 µm. Bottom: magnification of the damaged area; scale bar = 100 µm. (**C**) The percentage of lesion area was evaluated on muscle stained with HEB as represented in B. Data presented as scatter plots with mean ± SEM, * *p* < 0.05. One-way ANOVA followed by Holm–Sidak post hoc test. N = 8 per group. (**D**) Myofiber cross-section areas (CSA) were measured on the whole muscle section by using ImageJ software (https://imagej.net/ij/). Mean ± SEM, one-way ANOVA: NS. N = 8 per group. (**E**) Muscle fiber size distribution. Myofibers were classified in clusters according to their area. Chi-squared: NS. N = 8 per group. (**F**) Cumulative percentage of myofibers in clusters. (**G**) *Wfdc3* mRNA level was quantified by RT-qPCR in the IMEP model. Quantifications were normalized to *Rplp0*. Data are presented as boxplots. Kruskal–Wallis followed by Dunn’s post hoc test: NS. N = 8 per group. (**H**) Effect of DUX4 induction on levels of Hif1α pathway mRNAs *Hif1a*, *Pdk1* and *Vegfa* in the IMEP model. mRNA levels were quantified by RT-qPCR and normalized to *Rplpl0*. Data are presented as boxplots. Kruskal–Wallis followed by Dunn’s post hoc test: NS. N = 8 per group.

**Figure 6 ijms-25-03327-f006:**
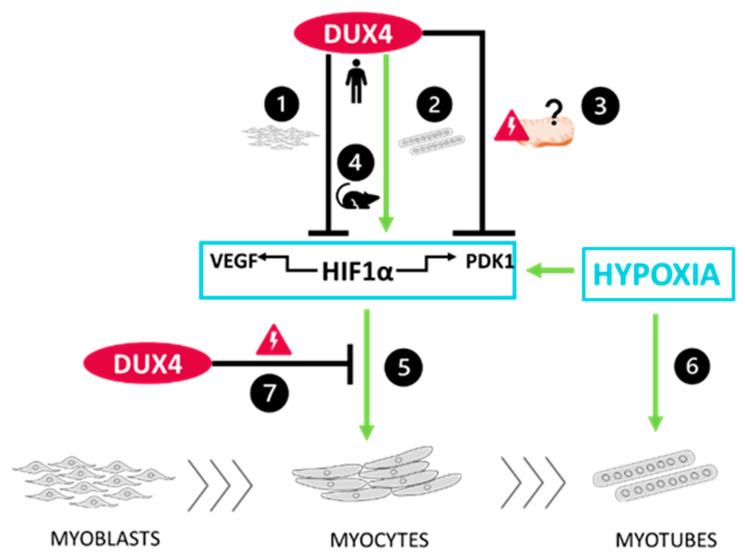
Schematic conclusions ❶ DUX4 inhibits HIF1α pathway in proliferating myoblasts but ❷ induces it in late differentiation into myotubes. Data regarding the two HIF1α target genes *VEGFA* and *PDK1* are consistent with those results. However, ❸ DUX4 decreases PDK1 protein levels regardless of the differentiation stage, likely due to DUX4-induced mitochondrial dysfunction. ❹ A DUX4–HIF1α axis also exists in mouse myoblasts as well as is adult muscle in vivo. Moreover, as we described in [40] in the context of adult myogenesis, hypoxia ❺ increases early myogenic differentiation in a HIF1α-dependent way and ❻ induces myocyte fusion independently of HIF1α. Finally, ❼ DUX4 represses HIF1α’s effects in early myoblast differentiation.

## Data Availability

Data are contained within the article and Appendix A.

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
