# Peer review of "The DUX4–HIF1α Axis in Murine and Human Muscle Cells: A Link More Complex Than Expected"

_ijms, 2024, doi:10.3390/ijms25063327_

Round 1

Reviewer 1 Report

Comments and Suggestions for Authors

The authors have studied the link of DUX4 and HIF1alpha in human and mice myoblast differentiation and muscle regeneration in vitro and in vivo. The study has wide implications in the feild of myogenesis and muscle regeneration. The authors have the tools of dissect the functional link between DUX4 and HIF1 alpha. However, there are a few questions (descibed below) before it can be considered for publication. In addition to following comments, authors must take in to consideration of general figure aesthetics such as font size across panels, proper scale bar and additional microscopy image adjustments. Here are my concerns:

1. Fig1.B,H, F, L: Authors must adjust the brightness-contrast levels in these panels for HIF1. Nuclei in red are not visible and weak signal appears as noise. Authors must show larger feild of views of these images and mention the cell count considered for quantification in the legend. At least 1000cells must be considered for such quantifications.

2. Fig1: have authors quantified the HIF nuclear intensities and mRNA levels across myoblasts, myocytes and myotubes in control and DUX4i cells? If the images of myoblasts, myocytes and myotubes for HIF1 expression were taken together as the same imaging parameters, then it appears from the images that in controls HIF1 increases from myoblasts to myocytes and then decresaes in myotubes. Interstingly, in DUX cells, there is no differences in HIF1 expression is significantly low in myoblasts and myocytes relative to myotubes. this may suggest that DUX4 cels are delayed in differentiation.

Do DUX4 cells for smaller myotubes than controls? what do fusion index analysis of myotubes between the two cell types suggest? It is an essential piece of information and must be included in results and figures. Authors can strengthen the HIF1 expression pattern in myoblasts, myocytes and myotubes of two cell types by qPCR as well.

2. FigS1F: merged and DAPI panels are swapped in DOX dosage of 15.6ng/ml. how many cells were taken into consideration for this panel? Authors are encouraged to show larger fields of views for these images as number of cells in all DOX panels appear signifiantly low than control suggesting toxicity. however cell toxicity assays suggest these concentrations of DOX are not toxic to the cells. Authors can then count the number of cells per feild of view in control and DOX treatments to rule out toxicity effects of DOX.

3. Fig2: western blot is missing for housekeeping gene (GAPDH, vinculin) in all western blot panels and authors must add it. I suggest authors compare myoblasts, mycytes and myotubes of each cells type as a group to understand mRNA and protein expression patterns of genes of interests such as PDK1 and VEGF.

4. Fig.4: Authors have used C2C12 cells and not primary mice myoblasts. These cells undergo slower differntiation and fusion  and are strikingly different relative to primary cells.  A genetically engineered cell lines exhibit clonal variability. How many iDUX4 clones of C2C12s were checked for similar phenotypes?

5. Fig4: How are authors sure that these cells at 1 and 2 days post seeding myoblasts are myocytes, respectively. Have authors checked expression of MyoD and MyoG? Why have authors not considered C2C12 cells derived myotubes for the study?

6. In the in vivo experiments the outcome of HIF1 downregulation in iDUX4 cells is muscle fibrosis. Have authors assessed the pool of total and activated SCs in these cells?

Reviewer 2 Report

Comments and Suggestions for Authors

The article The DUX4-HIF1α axis in murine and human muscle cells: a 2 link more complex than expected from Thuy-Hanh Nguyen et al. show that  the DUX4 and HIF1α 24 link differ according to the stage of myogenic differentiation is conserved between human and mouse muscle. The authors also show that HIF1α knock-down local expression exacerbated DUX4-mediated muscle fibrosis in a mouse model of DUX4. The authors suggest that the role of HIF1α in DUX4 toxicity is complex and that targeting HIF1α might be challenging in the context of FSHD therapeutic approaches.

Comments:

- Introduction: A lot of information that sometimes seems contradictory concerning the role of HIF1a in various contexts. Clarity according to timepoint could be improved on some points (aim of the article):

"HIF1α was identified as necessary for DUX4 toxicity by Lek et al in a genome-wide CRISPR-Cas9 screen performed to identify genes whose loss-of-function could allow survival of myoblasts expressing DUX4" (lines 84-86)

VS "Lek et al. showed that pharmacological HIF1α signaling inhibitors could improve DUX4-associated muscle phenotypes in FSHD-like zebrafish embryos" (lines 104-106) 

VS later in the paper the fact that siRNAs directed against HIF1a aggravate muscle lesion area by 10% (paragraph 2.7).

As highlighted in the discussion, the DUX4-HIF1a axis is indeed complex and difficult to clarify, despite the authors' efforts throughout the paper.

 Results : 

Paragraph 2. 3 : Clear difference in transcription of HIF1a target genes between myoblasts and myotubes but different protein expression for PDK1 gene in myotubes. No information regarding VEGF protein expression / It would be interesting to see if this is repeated for VEGF and what mechanism(s) would be altered or implemented in FSHD at the origin of this mRNA/Protein difference (if this is directly related to DUX4 or not. .).

Paragraph 2.4: Conservation of DUX4 effects in mice: observable for myoblasts but not so obvious for myocytes and chronologically different for myotubes/myofibers (next pararaph).

Paragraph 2.5: "Upon quantification of the damaged area, we found no statistical difference between the saline and the pCIneo control plasmid groups, therefore, we pooled data from both groups into a single control group." (lines 249-251).

Pooling the data could perhaps induce a bias (statistical analyses could be significant for the saline but not for the control plasmid, even if the latter are not significantly different from each other, you'd still have to compare one by one to be sure, as in Figure 4.

Titles moving from 2.1 to 2.3 then from 2.5 to 2.7 (absence of 2.2 and 2.6)
